# Human theca arises from ovarian stroma and is comprised of three discrete subtypes

Nicole Lustgarten Guahmich[1], Limor Man[1], Jerry Wang[1], Laury Arazi[1], Eleni Kallinos[1], Ariana Topper-Kroog[1], Gabriel Grullon[1], Kimberly Zhang [1], Joshua Stewart[1], Nina Schatz-Siemers[2], Sam H. Jones[1], Richard Bodine[1], Nikica Zaninovic[1], Glenn Schattman[1], Zev Rosenwaks[1] & Daylon James [1,3]✉

Theca cells serve multiple essential functions during the growth and maturation of ovarian follicles, providing structural, metabolic, and steroidogenic support. While the function of theca during folliculogenesis is well established, their cellular origins and the differentiation hierarchy that generates distinct theca sub-types, remain unknown. Here, we performed single cell multi-omics analysis of primary cell populations purified from human antral stage follicles (1–3 mm) to define the differentiation trajectory of theca/stroma cells. We then corroborated the temporal emergence and growth kinetics of defined theca/stroma sub-populations using human ovarian tissue samples and xenografts of cryopreserved/thawed ovarian cortex, respectively. We identified three lineage specific derivatives termed structural, androgenic, and perifollicular theca cells, as well as their putative lineage-negative progenitor. These findings provide a framework for understanding the differentiation process that occurs in each primordial follicle and identifies specific cellular/molecular phenotypes that may be relevant to either diagnosis or treatment of ovarian pathologies.

[1] Ronald O. Perelman and Claudia Cohen Center for Reproductive Medicine and Infertility, Weill Cornell Medicine, New York, NY 10065, USA. [2] Department of Pathology and Laboratory Medicine, Weill Cornell Medicine, New York, NY 10065, USA. [3] Department of Obstetrics and Gynecology, Weill Cornell Medicine, New York, NY 10065, USA. ✉email: djj2001@med.cornell.edu

The development of oocyte-bearing follicles within the human ovary is a dynamic process that requires communication between the oocyte, granulosa cells (GCs) and supporting cells from the ovarian stroma, commonly referred to as theca cells (TCs). This process begins with the activation of primordial follicles, which are comprised of an oocyte surrounded by a single layer of flattened GCs[1]. As the oocyte exits this dormant stage, GCs acquire a cuboidal morphology, begin to proliferate and together with the oocyte secrete signals that recruit surrounding stroma cells and promote their differentiation toward TC fate. Once formed, the theca has historically been believed to be comprised mainly of two morphologically distinct cell types: steroidogenic theca interna cells which produce androgens that are then metabolized into estrogen by GCs; and theca externa cells that provide structural support to the growing follicle[2]. In addition, as the follicle develops the TC layer is infiltrated by vascular and immune cells, becoming the main interface between the follicular unit and the rest of the organism. Although proper differentiation of stroma cells into specialized TCs is known to be essential for fertility, the mechanisms that drive TC specification and govern their interaction with follicle-resident cells during oocyte maturation are not well-defined.

In humans, the origin, differentiation trajectory and cellular heterogeneity that exist in the TCs surrounding the follicle remains particularly elusive. Unlike GCs, which are readily accessible in an IVF setting, isolation, and culture of TCs requires direct biopsy of pre-ovulatory follicles. In addition, most studies have relied on mechanical dissociation of the theca layer from isolated antral follicles or generation of TCs from isolated ovarian stroma cells[3,4], thereby yielding a heterogeneous mixture of stromal, theca, vascular and immune populations. Advances in single cell RNA sequencing (scRNASeq) can circumvent these limitations to provide high resolution snapshots of the transcriptional heterogeneity that exists within different compartments of ovarian follicles. These studies, when followed by marker identification and careful characterization of cellular heterogeneity, are a valuable resource for investigating the heterogeneity and provenance of TCs.

Recent work by our group[5] and others[6,7] used scRNASeq to rigorously define the diversity of cellular populations in the human ovary. Interestingly, these studies identified TCF21, a transcription factor that has been linked to fibroblast differentiation in lung, kidney and heart development[8], as uniquely expressed in ovarian theca/stroma. During embryonic gonadogenesis TCF21 has been shown to act downstream of SRY to suppress Nr5a1/Sf1 expression[9], and its deletion results in male to female sex reversal in mice[10]. Lineage tracing experiments in adult mice showed that Tcf21 cells give rise to Leydig cells, the male counterpart of theca interna cells in the testis, during normal aging and in response to injury[11]. Nonetheless the role and expression pattern of TCF21 in the adult ovary throughout follicle development remains to be established.

In this study we expanded on previously published results[5] to focus on elucidating the differentiation trajectory of ovarian TCs within human antral follicles, as well as the heterogeneity that exists within this population. Single-cell RNASeq and temporal immunophenotype analysis distinguished discrete cellular identities within the theca/stroma compartment and defined their origins over progressive stages of follicle development. Sequential incorporation of thymidine analogs ethinyl-deoxyuridine (Edu) and chloro-deoxyuridine (CldU) delineated the relative growth kinetics of cells within the TC differentiation trajectory and chromatin accessibility assay (ATAC-Seq) revealed reciprocal activity of TCF21 and NR5A1/SF1 in mediating the acquisition of structural versus androgenic TC fate, respectively. These results provide a comprehensive cellular and molecular analysis of TC

ontogeny over the course of human folliculogenesis. Defining the cellular origins and differentiation mechanisms of theca/stroma may shed light on the etiology of syndromes in which aberrant TC biology contributes to pathology, such as hyperthecosis[12] and polycystic ovary syndrome[13]. Moreover, control of TC identity will be vital to the development of "artificial ovaries"[14] and/or other methods for in vitro maturation of oocytes[15].

## Results

**Isolation and purification of theca/stroma cells with scRNA-Seq.** We have previously identified a panel of membrane-localized proteins that specifically distinguish GCs and theca/stroma populations[5]. Using this panel and a stepwise enzymatic dissociation protocol, we enriched for GC, theca/stroma and other ovary-resident cell types from antral follicles taken from freshly resected adult ovaries. We purified cell populations from a total of 13 antral follicles ($n = 3$ donors, see Materials and Methods and Supp Table 1), which were incorporated into 8 individual library preparations and submitted for scRNASeq and analysis. After routine quality control[16], 48,147 cells remained for downstream for analysis, with cells from follicles/donors contributing broadly to cell clusters (Supp Fig. 1a, b). Unbiased grouping revealed 22 distinct clusters (0–21, Supp Fig. 1a and c), which, upon exclusion of those comprised predominantly of cells with low number of features/counts (asterisks in Supp Fig. 1d, e and Materials and Methods) and determination of highly enriched markers within each cluster (Supp Fig. 1c and Supp Data 1), could be attributed to 6 major ovarian cell populations known to be present at the antral follicle stage: endothelial cells (ECs), smooth muscle cells (SMCs), GCs, ovarian epithelium, hematopoietic cells (Hem), and the largest coherent group of cells representing a theca/stroma cluster (Supp Fig. 1a).

We focused our analysis on 7 clusters representing theca/stroma (Seurat clusters #1, 2, 3, 4, 5, 11, 18), a total of 23,736 cells. Reclustering at default resolution (0.8, FindClusters) resolved 12 distinct Seurat clusters representing the theca/stroma subset (Fig. 1a), with clusters 3, 6 and 8 containing cells that were predominantly at S or G2/M phase (Fig. 1b). Among follicles for which corresponding GCs were also sequenced, we performed an analysis of transcripts that have been shown to be increased or decreased in atretic follicles[17] (Supp Fig. 1f). Of the follicles that included GCs in the library preparation, a single follicle (Pt2_1) displayed a signature that was consistent with the atretic phenotype. To correlate cluster allocations with their corresponding localization within the theca/stroma layer, we selected a minimal cohort of transcripts (Fig. 1c) that were enriched in closely associated clusters (dotted lines in Fig. 1a and Supp Data 1) and have been previously implicated in TC differentiation[18–20], and used RNA in situ hybridization (RNA ISH) to localize these transcripts in antral follicles (Fig. 1d–g). Transcription factor 21 (TCF21), a transcription factor known to be involved in fibroblast differentiation and to have a role in Leydig cell development[11], was significantly enriched in clusters 0, 2, 5, and 9 (Supp Data 1) and RNA ISH showed it to be expressed in a gradient within the theca/stroma layer (Fig. 2d), with the highest level observed in cells that were 4–5 cell layers away from the GC basement membrane. Patched 1 (PTCH1), the Hedgehog (Hh) signaling receptor, was significantly enriched in clusters 4, 6, and 7 (Supp Data 1) while enrichment of the inhibitory decoy receptor Hedgehog interacting protein (HHIP) was limited to clusters 4, 6, and 8 (Supp Data 1). Accordingly, RNA ISH revealed an increased zone of PTCH1 expression (Fig. 1e) relative to HHIP (Fig. 1f) within the theca/stroma layer, with both transcripts localized to cells immediately adjacent to the GC basement membrane. Also within the zone of PTCH1 expressing cells, Nuclear Receptor Subfamily 5 Group A Member

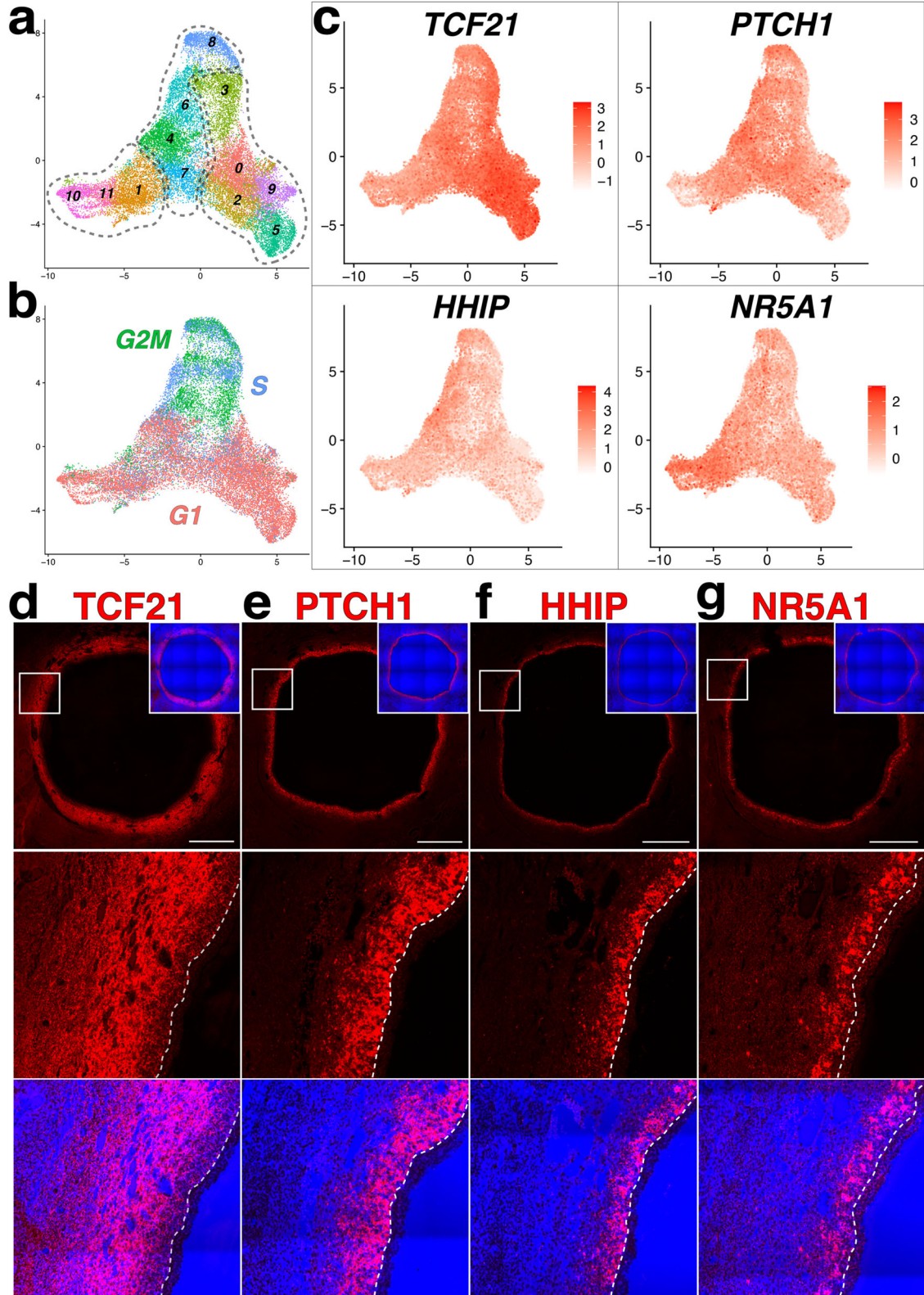

**Fig. 1 Identification of theca/stroma sub-types by transcript expression within the theca/stroma layer. a–b** UMAP plot of the isolated theca/stroma populations highlighting Seurat clusters (**a**) or populations colored by cell-cycle phase. G1 = pink, G2M = green and S = blue. **c** UMAP feature plots showing the expression of TCF21, PTCH1, HHIP and NR5A1 among theca/stroma cells. **d–g** RNA in situ hybridization signal for TCF21 (**d**), PTCH1 (**e**), HHIP (**f**), and NR5A1 (**g**) transcripts in antral follicles; probe signal is shown in red with the blue channel showing transmitted light. Magnified views are denoted by white boxes. Scale bars – 1 mm.

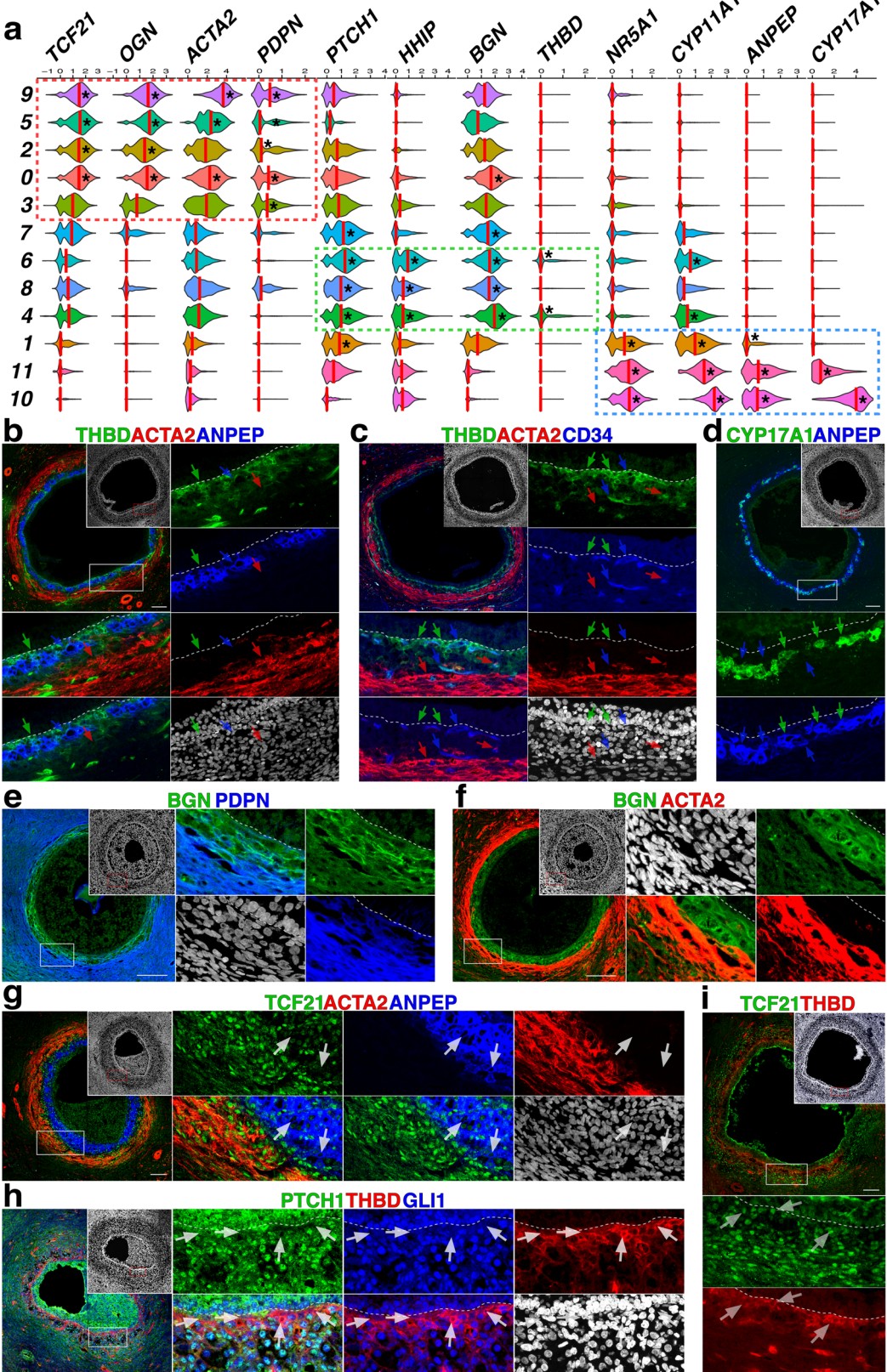

1 (*NR5A1*), a transcription factor associated with acquisition of steroidogenic cell fate[21], was observed in cells that were represented by Seurat clusters 1, 10, and 11 (Supp Data 1). Notably, *NR5A1*-expressing cells were not immediately adjacent to the GC basement membrane but localized in a pattern that appeared mutually exclusive with *HHIP* (Fig. 2g). Together, these 4 transcripts allowed us to distinguish anatomically distinct layers among the theca/stroma of antral follicles (dotted lines in Fig. 1a): GC basement membrane-associated *HHIP*+ and *NR5A1*+ cells residing within a *PTCH1*+ region that overlaps with a gradient of *TCF21*+ cells that demarcate the outer limit of the theca/stroma layer.

**Fig. 2 Immunophenotypic analysis of cell types within the theca/stroma layer. a** Violin plots of transcripts differentially expressed across theca clusters. Genes that were used to distinguish discrete lineage-specific sub-populations are enclosed within dotted lines: red = sTC, green = pfTC and blue = aTC. Red lines show median expression levels for all cells in that cluster; black asterisks show levels that are significantly enriched ($p$ value $\leq 1 \times 10^{-40}$.) **b** Representative immunolabeling of antral follicles for THBD, ACTA2 and ANPEP; arrows are color-coded and indicate position of the layer where identified cells are located. **c** Immunolabeling of antral follicles for THBD, ACTA2 and the EC marker CD34; arrows show representative cells: green = uniquely expressing THBD, blue = ECs that stain for both CD34 and THBD, and red = cells that are positive for ACTA2. **d** Immunolabeling for CYP17A1 and ANPEP, blue arrows show representative cells that stain for ANPEP, while green cells show ANPEP$^+$CYP17A1$^+$ cells. **e–f** Immunolabeling for BGN and PDPN (**e**) or ACTA2 (**f**). **g** Antral follicle stained for the transcription factor TCF21, ACTA2 and ANPEP; arrows indicate representative ANPEP$^+$ cells that show reduced TCF21. **h** Immunolabeling of an antral follicle for PTCH1, THBD and GLI1; white arrows indicate THBD$^+$ cells showing decreased expression of PTCH1 and GLI1. **i** Labeling of TCF21 and THBD in antral follicles; white arrows show THBD cells that are also positive for TCF21. Scale bars in (**b–i**) – 100 μm. Nuclear counterstain (DAPI) is shown in white in (**b–i**); white boxes in (**b–i**) indicate areas magnified in adjacent panels; white dotted line in (**b–f**) and (**h–i**) shows the GC basement membrane.

### Identification and characterization of multiple cell types within the theca/stroma layer.

We next elaborated upon the phenotype of distinct populations identified by clustering and RNA ISH using immunolabeling. First, we identified a panel of transcripts that were uniquely enriched among the theca/stroma clusters (Fig. 2a and Supp Fig. 2a). Clusters 0, 2, 3, 5, and 9 were significantly enriched for transcripts that have previously been associated with cell contractility/structural support (Smooth-Muscle Actin [ACTA2][22]) and differentiation/signaling of mesenchymal cells (TCF21[8,11], Podoplanin [PDPN][23], Osteoglycin [OGN][24]); we designated these cells as structural TCs (sTCs, framed in red in Fig. 2a). Clusters 4, 6, and 8, identified by expression of HHIP, were also enriched for Biglycan (BGN), a target of TGFb signal activation[25], and Thrombomodulin (THBD), a protein typically expressed on ECs[26]; because HHIP-expressing clusters identify cells directly adjacent to the GC basement membrane that are distinct from NR5A1$^+$ cells, we designated these as peri-follicular TCs (pfTCs, framed in green in Fig. 2a). Clusters 1, 10, and 11 expressed NR5A1 as well as Alanine Aminopeptidase (ANPEP), a surface marker of steroidogenic TCs[5], and the enzymes Cytochrome P450 Family 11 Subfamily A Member 1 (CYP11A1) and Cytochrome P450 Family 17 Subfamily A Member 1 (CYP17A1); given the essential role of CYP17A1 in producing androgens[27], we designated these cells as androgenic TCs (aTCs, framed in blue in Fig. 2a). Finally, cluster 7 exhibited significant enrichment of PTCH1 and BGN, but lacked expression of other markers of differentiation (e.g., ACTA2, THBD, CYP17A1); based on this we designated this population as a putative lineage-negative theca progenitor cell (Lin$^{neg}$TPC, framed in gray in Fig. 3a).

Immunolabeling corroborated the unique phenotypes that were identified by clustering (Fig. 2a) and localized them within the theca layer of antral follicles. Labeling with antibodies specific for THBD, ANPEP and ACTA2 (Fig. 2b) identified pfTCs immediately adjacent to the GC basement membrane (green arrows), aTCs (blue arrows) and sTCs (red arrows), respectively. Importantly, while Lin$^{neg}$TPCs could not be specifically identified here based on expression of surface markers, we have designated these as a putative cell that resides between the aTC and sTC layer, in THBD$^{neg}$ANPEP$^{neg}$ACTA2$^{neg}$ cells that express low TCF21 (Fig. 1d), high PTCH1 (Fig. 1e), and no HHIP (Fig. 1f). Co-staining for THBD with ACTA2 and the EC marker CD34 (Fig. 2c) confirmed that although some ECs closely associated with the follicular basement membrane expressed THBD (blue arrows), there were also CD34$^{neg}$ACTA2$^{neg}$THBD$^+$ pfTCs present (identified with green arrows). Coinciding with transcriptional levels (Fig. 2a), CYP17A1 expression was restricted to a subset of ANPEP$^+$ cells (green arrows), with ANPEP$^+$CYP17A1-$^{neg}$ cells (blue arrows) presumably identifying precursors to committed aTC fate (Fig. 2d). Immunolabeling of BGN (Fig. 2e, f) revealed that it is restricted to the inner theca layer, showing little

overlap with PDPN$^+$ cells (Fig. 2e) that are localized in the outer theca region and extend throughout the surrounding stroma. Similar exclusivity is observed between BGN and ACTA2 (Fig. 2f), with ACTA2$^+$ cells (stTCs) enveloping the BGN$^+$ region, but not extending throughout the stroma like PDPN$^+$ cells.

As predicted by the scRNASeq analysis of cells from antral follicles, TCF21 was enriched within the ACTA2$^+$ sTC layer (Fig. 2g), with its expression decreased in aTCs (denoted by ANPEP staining, white arrows). This coincides with RNA ISH data and suggests that while TCF21 plays a role in specification/development of the sTC layer, its expression is downregulated in cells that differentiate to aTC fate. Notably, TCF21 is observed in cells that reside between the ANPEP$^+$ and ACTA2$^+$ layers in putative Lin$^{neg}$TPCs (Fig. 2g) as well as in THBD$^+$ cells (Fig. 2i). As indicated by RNA ISH (Fig. 1e), PTCH1 staining was generally enriched in cells between the GC basement membrane and the sTC layer. PTCH1 is both a receptor that activates Hh signaling, as well as a target of pathway activation[28]; HHIP acts as a decoy receptor for Hh ligands, thereby having an inhibitory effect on activation[29]. Accordingly, PTCH1 and GLI1, both effectors of Hh signaling, were reduced in THBD$^+$ cells (Fig. 2h), suggesting that HHIP acts cell-autonomously in pfTCs to inhibit signal activation. Together, scRNASeq and immunolabeling enabled the identification of 4 distinct cellular populations within the theca layer of human antral follicles, with one of these populations representing a putative progenitor (Lin$^{neg}$TPC). The data also indicate an interplay between Hh signaling and TCF21/NR5A1 in the specification of sTC, aTC and pfTC identities.

### Delineation of population hierarchy with temporal immunophenotype analysis.

To define the order by which distinct cell populations emerge during folliculogenesis and the timing of TC differentiation, we immunolabeled follicles beginning at primordial up to early antral stages (Fig. 3). Beginning at primary to secondary stages, cells in the periphery of follicles showed positivity for ENG (red arrows in Fig. 3a.i–ii), ACTA2 (blue arrows Fig Fig. 3.i–ii) or both (red and blue arrows), thereby demarcating the initial specification of TCs. As follicles approach antral stage (Fig. 3a.iii), ENG and ACTA2 remain colocalized, however, a layer of ACTA2$^{neg}$ENG$^{neg}$ANPEP$^+$ cells is also evident. Few of the ANPEP$^+$ cells displayed a definitive androgenic phenotype, defined by CYP17A1 expression, at this stage (Fig. 3b), and acquisition of THBD$^+$ phenotype occurred in parallel to emergence of ANPEP$^+$ cells at late pre-antral stage (Fig. 3c). Specification of the theca precursors in primary follicles, marked by the presence of ACTA2$^+$ cells (Fig. 3d.ii, blue arrows) coincided with an increase in TCF21 positivity in cells at the periphery of the GC basement membrane (Fig. 3d.iii, green arrow). At secondary stage, ACTA2 and TCF21 were colocalized in the cell layer surrounding follicles (Fig. 3d.iv) and this was corroborated by RNA

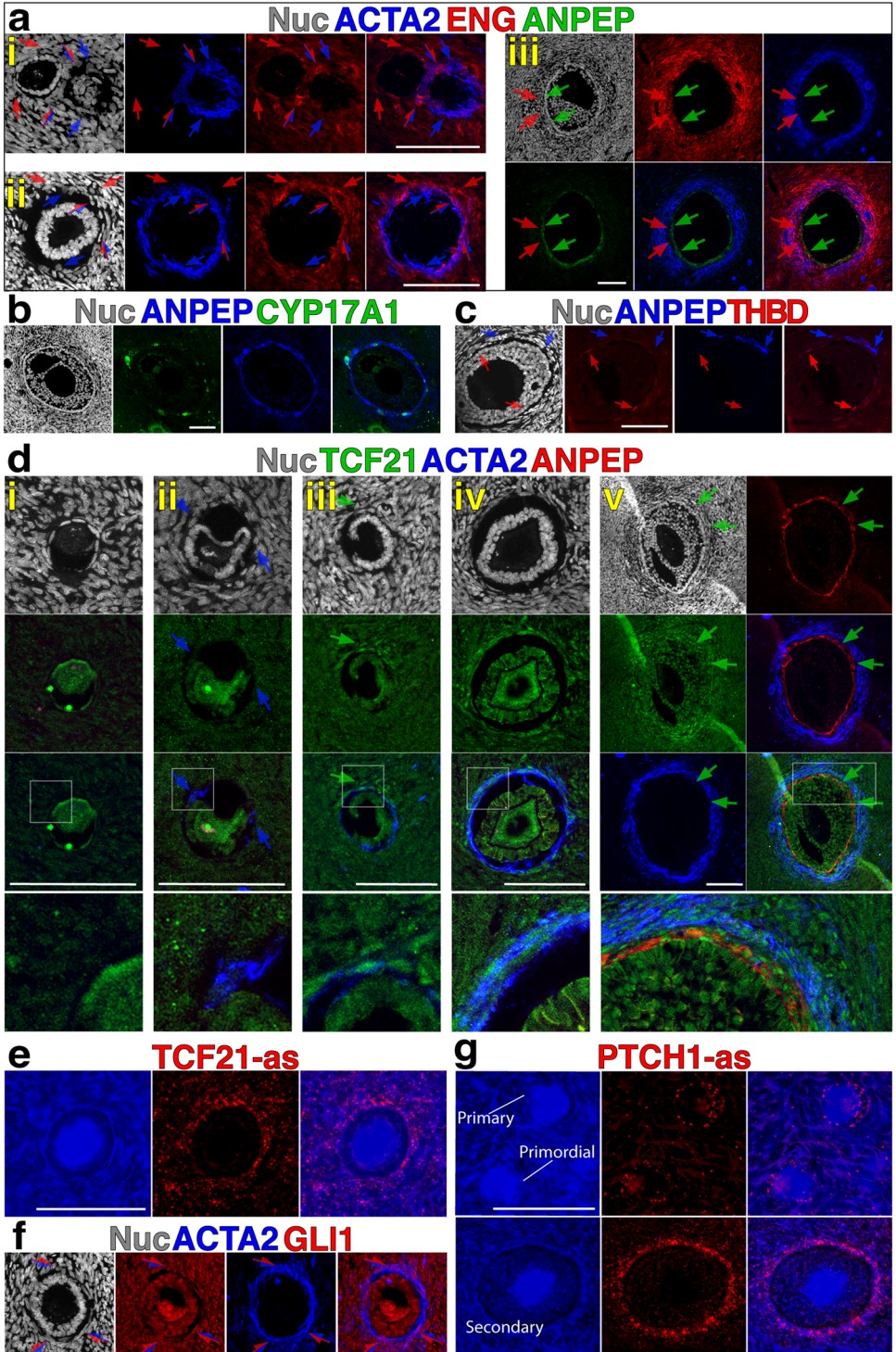

**Fig. 3 Immunolabeling and RNA ISH reveals progressive emergence of theca lineage markers. a** Immunostaining for ACTA2, ENG and ANPEP in primary (i), secondary (ii) and early antral (iii) follicles. Red, blue, and red/blue arrows point to cells that are positive for ENG, ACTA2 or both markers, respectively; in (iii) green arrows indicate ANPEP+ cell layer, while red arrows indicate an ENG+ area that is negative for both ANPEP and ACTA2. **b–c** Staining of early antral follicles for ANPEP and CYP17A1 (**b**), or ANPEP and THBD (**c**); red arrows indicate THBD+ cells and blue arrows show ANPEP+ cells. **d** Representative Immunolabeling for TCF21, ACTA2 and ANPEP in (i) primordial, (ii) transitioning primary, (iii) primary, (iv) secondary and (v) early antral follicles. Blue arrows indicate ACTA2+ cells and green arrows indicate TCF21+ cells; white boxes indicate areas magnified in bottom panels. **e** RNA ISH labeling for TCF21 RNA in a primary follicle. **f** Staining for ACTA2 and GLI1 in a secondary follicle; representative cells expressing both markers are indicated with red and blue arrow. **g** RNAScope labeling for PTCH1 RNA in primary and secondary follicles. Blue channel shows transmitted light in (**e** and **g**). Scale bars – 100 μm. **a–e** Nuclear counterstain (DAPI) is shown in white in (**a–d** and **f**). as = anti-sense RNA probe in (**e** and **g**).

ISH (Fig. 2e). Notably, at the early antral stage (Fig. 3d.v), the ACTA2$^{neg}$ANPEP$^+$ cell layer was also positive for TCF21, unlike ANPEP$^+$ cells at later antral stages which show reduced TCF21 levels (Fig. 2g). In cells of secondary stage follicles that express TCF21 (Fig. 3e), nuclear localization of GLI1 was observed in the ACTA2$^+$ region (Fig. 3f), suggesting activation of Hh signaling; but although *PTCH1* was enriched in TCs in the periphery of follicles at pre-antral stages, it was enriched in GCs at primary stages (Fig. 3g). Taken together, immunolabeling and RNA ISH for these key factors demonstrated that emergence of theca, defined as a specialized cell proximal to the activated follicle that shows a distinct phenotype from the surrounding stroma, can be identified as early as primary stages. Initially these cells show shared expression of ACTA2 and ENG as well as enrichment for TCF21 and PTCH1. Observation of this cell phenotype preceded the mutually exclusive emergence of ANPEP$^+$ and THBD$^+$ cells at early antral phases, suggesting that it may represent a founder population of Lin$^{neg}$TPCs identified in antral follicles.

**Quantification of proliferation kinetics in ANPEP$^+$ and THBD$^+$ populations.** In antral follicles, scRNASeq analysis of TCs in antral follicles identified three clusters comprised predominantly of proliferative cells that were linked to pfTC and sTC phenotype, and immunolabeling of antral follicles suggested that KI67$^+$ proliferative cells were relatively infrequent among ANPEP$^+$ cells and enriched in THBD$^+$ cells (Fig. 4a). To provide a more refined measure of cell proliferation dynamics, we injected xenograft bearing mice with sequential injection of thymidine analogs EdU and CldU, as previously described[30,31]. After 11 weeks, with weekly MRI monitoring beginning at 8 weeks to detect incipient antral follicle emergence, growth was observed and animals were consecutively injected with EdU and CldU in advance of xenograft recovery (Fig. 4b). EdU was injected at hours 0 and 24, followed by CldU injections at 48 and 72 h. Twenty-four hours after the last injection, the tissue was recovered and processed. Quantification of proliferating cells in the CYP17A1$^+$, ANPEP$^+$, and THBD$^+$ fractions revealed significantly different values (Fig. 4c–f and Supp Data 2). The THBD$^+$ population had the highest proliferative index, and notably, the degree of difference between ANPEP$^+$ and CYP17A1$^+$ populations was reduced in the measure of actively (Ki67, Fig. 4e) versus ancestrally (EdU, Fig. 4d) proliferating cells, suggesting that the rate of proliferation within ANPEP$^+$ derivatives is reduced as they arrive at a terminal CYP17A1$^+$ androgenic fate (Fig. 4f). Notably, in early follicles ($n = 45$), including primordial thru early secondary stages (i.e., <5 layers of GCs, Fig. 4g), cells in the periphery of follicles proliferated at an increased rate relative to GCs (Fig. 4h). Combined with temporal immunolabeling (Fig. 3), these results suggest that activation and proliferation of stroma/theca is a hallmark of early follicle development.

**Variable chromatin accessibility at core transcriptional regulators NR5A1 and TCF21.** To assess chromatin changes that might be related to differentiation in our populations we purified and pooled the same TC (ANPEP$^+$ or CD55$^+$) and GC (CD99$^+$) populations from a single antral follicle for combined single nucleolar (sn)-RNASeq and scATACSeq. After quality control and joint RNA expression and chromatin accessibility analysis, 5024 remaining cells could be grouped into 16 distinct clusters that were attributed to 4 non-theca cell types using known markers (Fig. 5a, b and Supp Fig. 3): ECs (*CDH5*), GCs (*AMH*), hematopoietic cells (*PTPRC*), and SMCs (*RGS5*). Remaining cells were contained in larger cloud comprised of 10 Seurat clusters (Fig. 5c) that could be grouped according to similar patterns of expression as noted for cells analyzed by scRNASeq

(Fig. 2a). Theca/stroma clusters were further consolidated into four groups by similar parameters as those used to group clusters obtained by scRNASeq: putative Lin$^{neg}$TPCs (TPCs), sTCs, pfTCs, aTC precursors (pre-aTC) and definitive aTCs (Fig. 5d). We focused on the two transcription factors that were identified to be differentially regulated as the cells take on specialized fates: *TCF21* and *NR5A1*. Although *TCF21* expression was increased in Lin$^{neg}$TPCs and sTCs, chromatin accessibility at the *TCF21* locus was similar between all cell groups (Fig. 5e). In contrast, both expression of *NR5A1* and chromatin accessibility at the *NR5A1* locus were increased in pre-aTCs and aTCs (Fig. 5f). Resolution of peaks for each of these genomic loci revealed that despite *TCF21* accessibility remaining relatively similar between groups, a peak is evident in pfTCs, pre-aTCs and aTCs that is reduced in Lin$^{neg}$TPCs and sTCs (Fig. 5g), potentially demarcating a repressive influence. Conversely, multiple peaks are differentially distributed at the *NR5A1* locus (Fig. 5h), all of which suggest increased accessibility of enhancer regions in pre-aTCs and aTCs. Together, the combined gene expression and chromatin accessibility assay identify epigenetic mechanisms that govern acquisition of specialized TC fate.

**Discussion**

The theca that surrounds growing follicles, along with GCs, is a critical modulator of oocyte maturation, response to extrinsic stimuli, and ultimately, ovulation[32]. Unlike GCs, which exist within each primordial follicle in a dormant state and undergo extensive proliferation following activation[1], the cells that comprise the theca layer are induced from the surrounding ovarian stroma via factors that are secreted by GCs/oocytes of activated follicles. Theca cells have been distinguished, mostly by morphological criteria, into two broad categories: theca externa, which provides structural support; and theca interna, which mediates the steroidogenic function of the theca, generating androgens that serve as a substrate for aromatase activity in GCs. Although developmental origins of theca have been described in mice[19] and previous studies have isolated in vitro explants of bovine[33] and human[3,4] theca layers, the cellular origins of TCs within the ovary, as well as their differentiation hierarchy, remain poorly understood. Here, we have used a combination of primary ovarian tissue and ovarian cortical xenografts, along with scRNA, snRNA, and scATAC sequencing, to plot the differentiation trajectory and growth kinetics of theca/stroma cells during human folliculogenesis. Molecular deconstruction of the specialized cell types that reside within the theca/stroma layer may provide insight into ovarian pathologies that have a TC component (e.g., PCOS, hyperthecosis), and delineation of the differentiation hierarchy that gives rise to theca sub-types may facilitate strategies aiming to reconstruct the follicular milieu for in vitro maturation of oocytes or to generate an "artificial ovary".

Mouse studies have established that specification of TCs is dependent on GC-derived factors, specifically Indian hedgehog (Ihh) and/or desert hedgehog, which are secreted in response to Gdf9 from the oocyte upon follicle activation[19,20]. Developmentally, theca have been shown to arise from two progenitor sources: Wilms tumor 1 (Wt1)$^+$ cells that originate from gonadal primordium, and Gli1$^+$ cells that migrate from the mesonephros to colonize the gonad[19]. Yet the molecular phenotype and/or resting place of these cells in the postnatal gonad have not been identified, nor has the differentiation trajectory of theca subtypes during adult folliculogenesis been charted. Using scRNASeq and RNA and protein immunolabeling we were able to identify and localize 3 lineage specific sub-types within the TC of antral follicles: sTCs, aTCs, and previously undescribed pfTCs. Moreover,

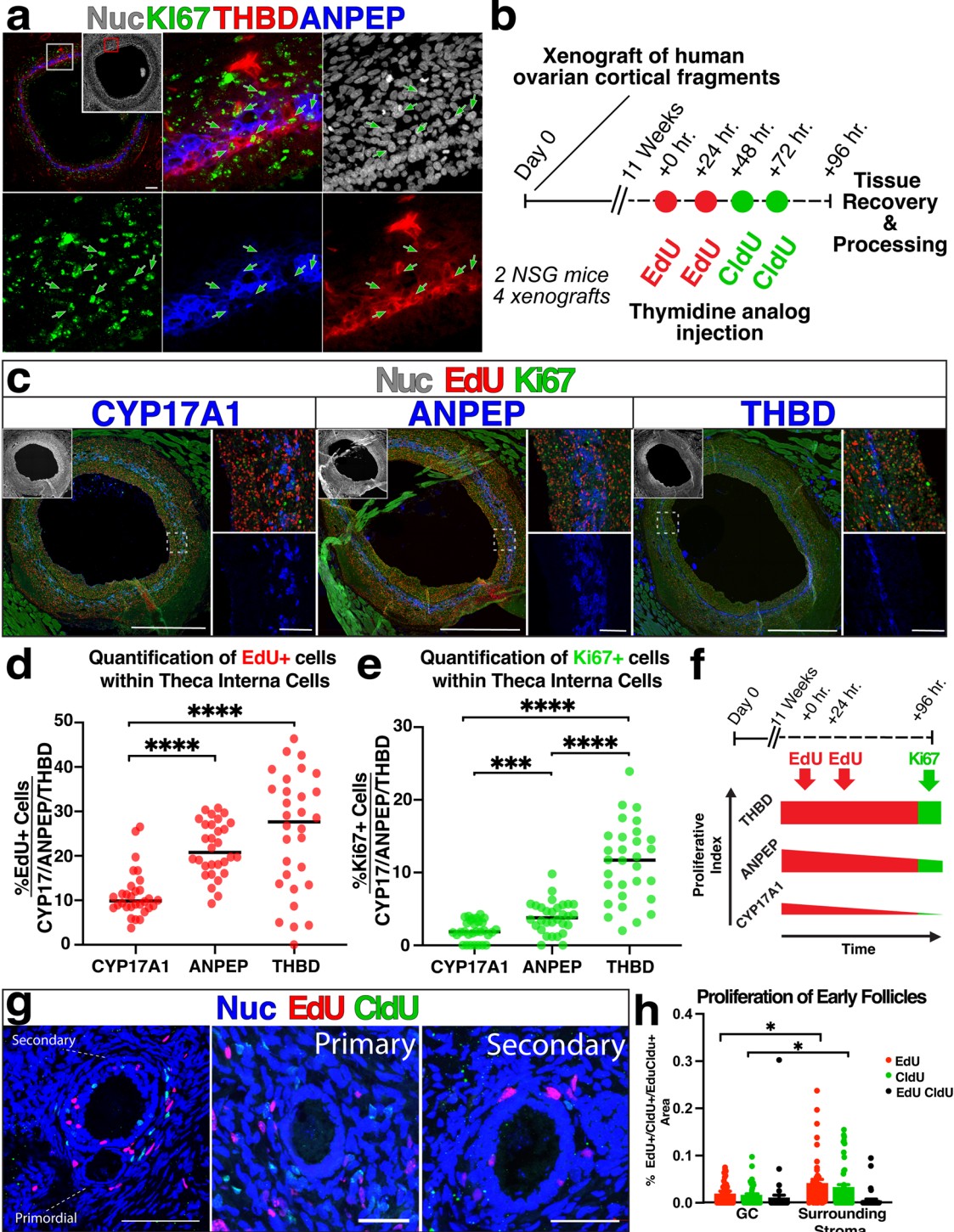

**Fig. 4 Mitotic index of THBD+, ANPEP+ and CYP17A1+ reveals decreased proliferation with acquisition of androgenic fate. a** Representative staining for Ki67, THBD and ANPEP in antral follicles; green arrows indicate Ki67+ cells. **b** Schematic of experimental approach for sequential labeling with thymidine analogues; red and green dots indicate times of EdU and CldU injections, respectively. **c–e** Xenografted antral follicles labeled to detect EdU incorporation (red) ki67 (green) and either CYP17A1, ANPEP, or THBD as indicated (blue). Cells within the CYP17A1/ANPEP/THBD compartment that stained positive for EdU (**d**) or Ki67 (**e**) were quantified. **f** Schematized representation of proliferation among sub-populations in antral follicles. **g** Representative images of ovarian xenograft tissue containing pre-antral follicle stages immunolabeled for EdU (red) and CldU (green). **h** Quantification of proliferative cells within the GC and theca compartment of early follicles (primordial, primary, secondary <5 GC layers), as measured by incorporation of EdU (red bar), CldU (green bar) or both (black bar). Nuclear counterstain (DAPI) is shown in white in (**a** and **c**) and in blue in (**g**). Scale bars: in (**a**) – 100 µm; in (**c**) large panels 1 mm, small panels 50 µm; in (**g**) left panel – 100 µm, Primary and Secondary panels – 50 µm;. *p* values *<0.05, **<0.005, ***<0.0005, ****<0.00005.

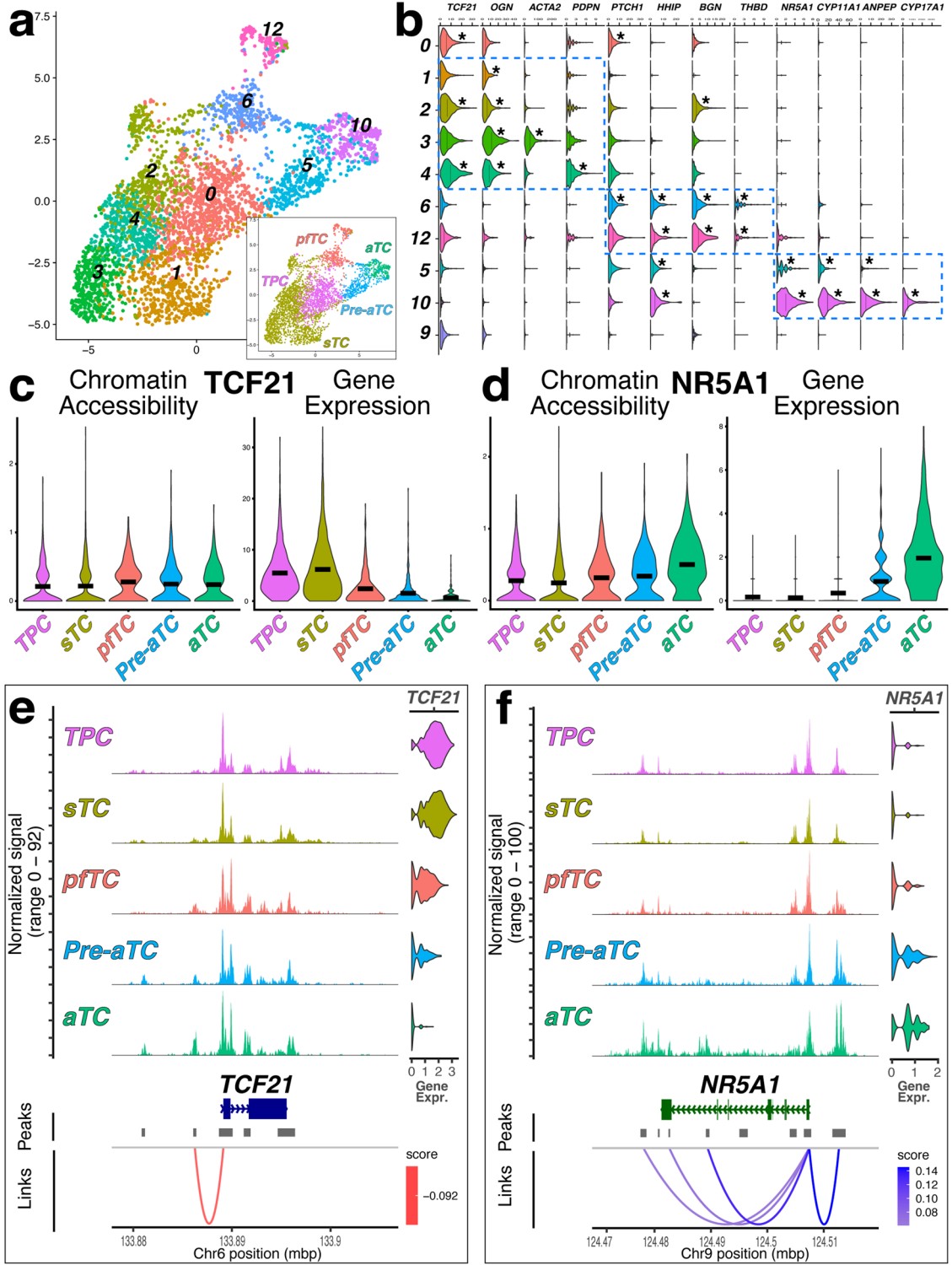

**Fig. 5 Combined snRNASeq and scATACSeq reveals reciprocal accessibility of the NR5A1 and TCF21 genomic loci as cells differentiate.**
**a**–**b** Designation of theca/stroma subclusters based on snRNASeq using previously identified referents; UMAP graphs in (**a**) show Seurat clusters with groups redesignated in the inset based on markers of theca subpopulations shown in violin plots in (**b**); each population is demarcated by dotted blue lines.
**c**–**d** Violin plots showing overall chromatin accessibility and RNA expression by theca/stroma subtype for the TCF21 (**c**) and NR5A1 (**d**) locus.
**e**–**f** Sequencing tracks, snRNA Expression data and predicted regulation of chromatin accessibility peaks to gene expression for the TCF21 (**e**) and NR5A1 (**f**) locus, by theca subtype. Positive interactions are depicted in blue and negative interaction are depicted in red. **b**–**d** Lines in violin plots indicate mean expression level; asterisks indicate significant enrichment of a transcript relative to all other Seurat clusters ($p$ value $\leq 1 \times 10^{-43}$).

we describe a putative Lin^negTPC that has not been clearly characterized but accounts for all 3 lineage specific subtypes.

Ovarian follicles are unique among adult systems in that they undergo a complex differentiation process that is more akin to developmental organogenesis than it is to organ/tissue homeostasis. Although NR5A1 has been well-described as a master regulator of steroidogenic function in TCs[34–36], transcription factors that initiate and/or modulate the specialization of other theca sub-types remain undefined. Previous work from our own group[5] and others[6] identified a TCF21-expressing population within the TC compartment, but here we have elaborated on the role of these cells within the hierarchy of theca/stroma differentiation. Specifically, we show that increased expression and nuclear localization of TCF21 coincides with activation of Hh signaling and specification of Lin^negTPCs and is attenuated with differentiation to aTC fate. TCF21 has been shown by many studies to play an important role in the specialization of numerous mesenchymal populations during development, including the heart[37], lung[38] and kidney[39]. In the heart, TCF21 is necessary for endothelial-to-mesenchymal transition that generates progenitors of cardiac fibroblasts[37]. Indeed, TCF21-expressing cells in the epicardium are multipotent and loss of TCF21 results in disproportionate differentiation to SMCs[40]. TCF21 null mice also die shortly after birth with hypoplastic lungs and kidneys that fail to develop alveoli and glomeruli, respectively[41]. Male mice null for TCF21 also show feminized genitalia, and this phenotype has been shown to stem from its negative regulation of NR5A1 that is necessary to dictate male gonadal cell fate during sexual differentiation[10]. This aligns with the reciprocal roles of TCF21 and NR5A1 at the initiation and terminus of TC differentiation that we observed and is supported by the relative gene expression and accessibility of these loci revealed by ATAC-Seq analysis. Finally, Shen et al. recently demonstrated that TCF21^+ cells represent biopotential progenitors in mouse fetal testis and ovary, and in testis, these cells generate Leydig and myoid cells and maintain testis homeostasis during aging or following injury[11]. Given the developmental equivalency of Leydig and TCs, these findings align with the expression pattern of TCF21 we observed over the course of theca/stroma differentiation.

It has long been appreciated that TCs perform multiple specialized functions over the course of folliculogenesis and are comprised of multiple sub-types that are morphologically distinguishable. However, the heterogeneity of the theca/stroma layer, along with its investment with vascular and immune cells, has made delineation of cellular origins and/or fate potential a challenge. Previous work has demonstrated ex vivo potential for ovarian stroma to form androgenic cells[3], however, the founder population in these studies was not defined and phenotypic milestones that demarcate differentiation toward diverse theca identities (e.g., externa versus interna) were not described. The differentiation hierarchy described here (Fig. 6) encompasses a broader and more detailed scope of theca/stroma differentiation, with clear phenotypic milestones that demarcate known TC subtypes that mediate structural (sTC) and steroidogenic (aTC) functions, as well as a previously undescribed population (pfTCs) that is closely associated with the GC basement membrane and vascular cells. Clear phenotypic segregation of the cell types that comprise the theca will advance efforts to recapitulate the cellular and molecular milieu ex vivo by ensuring an optimal balance of paracrine factors for cultivation of pre-antral stage follicles or immature oocytes in vitro; such optimization may improve the success rates of preantral follicle culture or in vitro oocyte maturation. Moreover, elucidation of molecular criteria that distinguish theca subtypes (including the previously undescribed pfTC population) may provide discrete

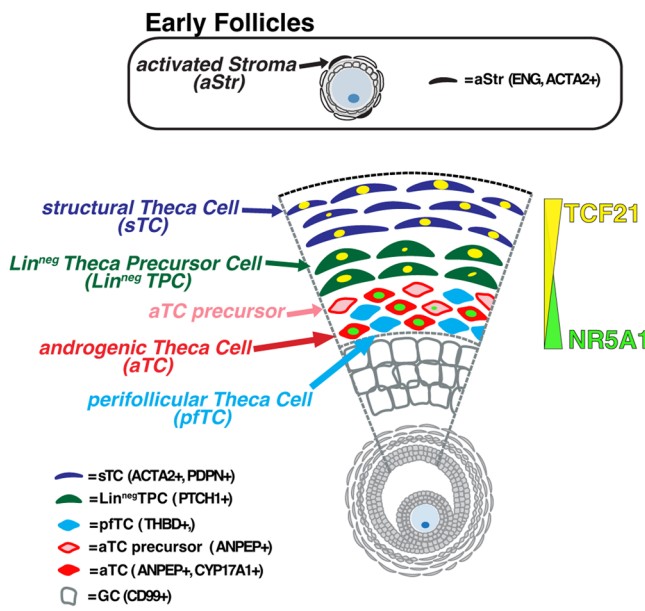

**Fig. 6 Proposed model of Theca differentiation.** In early growing follicles, including primary and secondary stages, activated (ENG^+, ACTA2^+) cells are identifiable near the follicular basal membrane. These cells differentiate to the multiple cell lineages observed within the theca layer of antral follicles, which include sTCs (ACTA2^+, PDPN^+, TCF21^+/^neg), aTCs (ANPEP^+CYP17A1^+/^neg), and pfTCs (ANPEP^negTHBD^+) as well as Lin^neg TPCs (PTCH1^+Lin^neg) with multilineage potential. Differentiation along this trajectory is driven by a reciprocal expression pattern of the transcription factors TCF21 and NR5A1, which govern specification of sTC and aTC identity, respectively.

measures/targets in the diagnosis/treatment of normal/pathological folliculogenesis.

The pfTC population emerged in parallel to ANPEP^+ cells at pre-antral stages. This, along with their localization in proximity to both GCs and vascular cells, suggests an important role in governing folliculogenesis, yet the function of THBD has not been examined specifically in the context of the ovary. Shen et al. observed expression of *Thbd* in non-endothelial Tcf21^+ cells in the testis that share a close transcriptional signature with Leydig cells[11], however the function of these cells has not been determined in testis or ovary. Mice null for Thbd are embryonic lethal before e9.5[42], and most studies of THBD function relate to its role in thrombosis[26], hence no data has been published examining ovary-specific deletion of THBD. Yet, thrombin, a cofactor of THBD, is present in follicular fluid[43,44] and has been implicated in mediating signaling events during ovulation unrelated to coagulation[45]. Indeed, in an isolated clinical case, a patient with a novel homozygous variant of THBD that caused a bleeding disorder presented with ovarian hemorrhage[46]; while existing animal studies and clinical observations are not sufficient to elaborate on the role of THBD in ovarian function, they support the hypothesis that ordered differentiation and distribution of THBD ^+ cells within follicles is vital. Our observation that HHIP was enriched in the pfTCs and coincided with reduced levels of PTCH1 and GLI1 (Fig. 4f, g) suggests that inhibition of Hh signaling is a pre-requisite for differentiation. Previous work showed that constitutive activation of hedgehog signaling specifically in the ovary (Amhr2-Cre/SmoM2) results in failure to ovulate, with oocytes "trapped" in peri-ovulatory follicles[47]. Collectively, these studies support a link between THBD/pfTCs and fertility/ovarian function; the molecular/phenotypic characterization of pfTCs, as well as their placement within the differentiation hierarchy and

spatiotemporal localization during human folliculogenesis, will help to define that link.

Many markers we have described (e.g., NR5A1, CYP11A1, CYP17A1, PTCH1) have established roles in the ovary, but the functional relevance of others (e.g., OGN, BGN, ACTA2) has not been clearly determined. OGN encodes for a leucine-rich glycoprotein that is involved in myriad processes including extracellular matrix (ECM) remodeling, collagen fibrillogenesis, and cell proliferation[48–50]. Interestingly, OGN has also been found to inhibit Hh-mediated differentiation of mesenchymal progenitors into SMCs in the developing gut[24]. Hence, OGN regulation of Hh may similarly influence theca/stroma differentiation. BGN is another leucine-rich glycoprotein that is localized to ECM, interacts with collagens, and acts as a signaling molecule during inflammation[51–54]. Finally, while ACTA2 is typically observed in perivascular cells and is commonly associated with contractility, it has also been documented in the ovary[55]. Similar to ACTA2 expression that has been noted in myofibroblasts in the heart[56], early expression of ACTA2 in incipient TCs surrounding the developing follicle could identify an early progenitor-like cell with phenotypic plasticity.

While advances in the treatment of female infertility have made tremendous gains over a relatively short time-period, the scope of the condition has also expanded, and traditional pharmacologic measures are often insufficient to address the root causes and/or manifestations of disease. This stems in part from incomplete understanding of the complex cellular and molecular dynamics that govern the growth and specialization of the cells that comprise ovarian follicles. We have combined single-cell multi-omics with rigorous descriptive assessment of human native and xenografted follicles to decipher the differentiation hierarchy of TCs, from activated stroma to three distinct terminal lineages present in early (≤3 mm) antral stage follicles. Deconstructing this process is particularly challenging for the female gonad, as access to human samples is limited and even in animal models, there are no in vitro systems that functionally recapitulate ovarian folliculogenesis like there are for spermatogenesis. As such, conclusions drawn from experiments using a limited amount of patient tissue (Fig. 4 EdU/CldU tracking) must be corroborated within the native ovary and/or across a broader distribution of patient/donor material. Even so, these methods fall short of the gold-standard for lineage tracing: genetically modified animal models that enable longitudinal cell-intrinsic labeling and tracking. Accordingly, the approach cannot provide a definitive a measure of in vitro lineage potential of TC sub-types or putative Lin[neg]TPCs. Yet the scale and physiology of human folliculogenesis is markedly distinct from animal models that are amenable to genetic lineage tracing (mouse, rats), and in vitro monolayer culture of theca and/or stroma is unlikely to recapitulate the physiological cues that are present in a three-dimensional growing follicle. Hence, while these data are limited, they offer basic insight into human ovarian biology and provide an important resource that may ultimately be applied toward: measurement of follicle quality; diagnosis and/or targeting of ovarian pathologies; generation and/or cultivation of ex vivo follicle; and/or in vitro maturation.

## Methods

**Materials & allocation**. Human ovarian material was obtained from two sources: *Braindead organ donors (Don) or Fertility Preservation Patients (Pt)*. For *organ donors* whole bilateral ovaries were obtained from brain-dead organ donors with consent, as a part of a collaboration with the International Institute for the Advancement of Medicine. Ovaries were resected and placed in sterile Leibowitz medium (Gibco) on ice for transport to the research laboratory for processing following a cold ischemic interval of ≤4 h[57]. Tissue obtained from donors was used for: antral follicle cell isolation for multi-omics studies, ISH/immunolabeling and cortical freezing for human ovarian xenograft experiments. For *Fertility*

*Preservation Patients*, we obtained discarded material from patients with informed consent and with approval of the Weill Cornell Institutional Review Board—Study title: Low temperature preservation of ovarian tissue; Study number: 0803009702. Discarded material was transported to the laboratory in sterile Leibowitz medium (Gibco) on ice following a cold ischemic interval of ≤2 h. Material obtained from Fertility preservation patients was used exclusively for antral follicle cell isolation for multi-omics studies. In total we obtained material from 6 different patients, 3 Tissue Donors and 3 Fertility Preservation patients. A detailed breakdown of human ovarian material and their use is included in Supplemental Table 1.

**Ethical approval for use of tissue**. All experiments using human tissue samples from either patients or organ donors were reviewed and approved by the Ethics Committee of the Institutional Review Board of Weill Cornell Medical College (IRB no. 0901010165). Written consent has been obtained from each patient or subject after full explanation of the purpose and nature of all procedures used.

**Isolation and dissociation of ovarian follicles**. These procedures were previously described in[5]. Briefly, antral follicles were resected from the cortex of whole ovaries obtained from tissue donors ($n = 2$) or from discard material from fertility preservation patients ($n = 3$). Only follicles deemed to be healthy by morphological appearance, identified by presence of a clear antrum, symmetric edges, and observable vascular network in the surface, and ≤4 mm in size (prior to selection of dominance[58]) were individually resected and placed in one well of a six-well dish containing cold DPBS (Gibco). After washing, the DPBS was removed and the follicles were bisected in 1 mL of Accutase (Invitrogen), followed by a 10 min incubation in a humidified incubator at 5% $CO_2$ and 37 °C. Immediately following the incubation, the Accutase was neutralized with 1 mL of DMEM/F12 media with 10% Knockout Serum Replacement and 1× Penn/Strep (Gibco), and the cells were dissociated from the tissues via repeated recycling of medium over bisected follicles with a P1000 micropipette. Media was collected and passed through a 100 μm filter (Corning), followed by centrifugation and aspiration of the supernatant to obtain a cell pellet (after this step the media contains mostly the GCs). After flushing of the follicle, the remaining tissue was placed in Collagenase (100 U/mL) and Dispase (1 U/mL) in Hanks balanced salt solution and incubated for 30 min in a humidified incubator at 5% CO2 and 37 °C.Tissue was recovered and triturated via pipetting through a P1000 micropipette tip and filtered through a 100 μm filter before centrifugation, supernatant aspiration, and labeling of cell pellets (this cell fraction includes mostly TCs). Single cell suspensions were immunolabeled and sorted prior to genomic studies.

**Labeling and FACS of cells from antral follicles**. These procedures were previously described in[5]. Cell pellets were washed with DPBS (Gibco) and resuspended in blocking solution containing either antibodies that were directly conjugated to CD99 and CD45 for the cells obtained from initial Accutase dissociation, or CD55 and ANPEP for the cell pellet obtained following Collagenase/Dispase dissociation. After incubation for 10 min at 4 °C, cells were washed, centrifuged, and resuspended in FACS buffer containing DAPI. Cells were run on a FACSJazz (BD) with collection and validation of the obtained fractions following initial sort. Purified cell population were washed, centrifuged, and submitted for either scRNASeq or combined snRNASeq/scATACSeq. In addition, for the investigation of the THBD population a fraction of the cell pellet obtained from the secondary Collagenase/Dispase dissociation was stained for CD34, THBD and ANPEP. Following cell sorting fresh single cell suspensions containing GCs (CD99+CD45[neg]) or TCs (CD55+ANPEP[+/neg]) were submitted for single-cell RNA sequencing(scRNASeq) or Combined Single-nucleolar RNA sequencing(snRNASeq) and Single-cell Assay for transposase-accessible Chromatin (ATAC-Seq) with high-throughput sequencing.

**Single-cell RNA sequencing (scRNASeq) and analysis**. The data described in this manuscript is listed on GEO under record number GSE192722.

These procedures were previously described in ref. [5]. Library preparation was performed using the Chromium Single Cell 3′ Reagent Kit and sequenced on a NovaSeq 6000 sequencer with 300 cycles per run. Samples were aligned to the human GRCh38 reference assembly using STAR aligner and downstream analysis was performed using the R package Seurat, v2.2. Details of bioinformatic analysis in contained within the script accompanying the manuscript, but briefly, to exclude poor quality cells, only cells with >200 but <4000 features and <20% of reads mapping to mitochondrial genes were retained. Counts were normalized using default normalization (Function NormalizeData). Retained cells from all libraries were integrated using the FindIntegrationAnchors and IntegrateData functions in Seurat and were then scaled to regress cell-cycle effects. To correct for patient-effects, the mutual nearest neighbor function was used. Function FindAllMarkers performed differential expression analysis between cells in a cluster relative to the remaining cells in the dataset, or alternatively, directly compared two populations. All transcripts with $p$ value of 0.005 or less were included in gene lists.

**Combined single-nucleolar RNA sequencing(snRNASeq) and single-cell assay for transposase-accessible Chromatin (ATAC-Seq) with high-throughput sequencing and analysis**. For scATACSeq libraries were prepared according to

the Chromium Next GEM Single Cell Multiome GEM Kit (CG000338 Rev E) and sequenced on a NovaSeq 6000 sequencer at an average depth of 20,000 reads per cell. Raw sequencing reads were aligned to GRCh38 reference genome using the *cellranger-atac count* pipeline and downstream analysis was performed using the R package Signac v1.5.0 and Seurat v4.0 on R v4.0.3. Details of bioinformatic analysis in contained within the script accompanying the manuscript. In summary, for quality control, cells were filtered using the following parameters: 1000 < nCount_RNA < 25,000, 1000 < nCount_ATAC < 10,000, nucleosome_signal <2 and TSS.enrichment >1. For peak calling the CallPeaks function was used and only peaks in standard chromosomes were analyzed. Joint UMAP visualization was built using the FindMultiModalNeighbors functions, and cell types were annotated based on RNA expression of curated markers. Chromatin accessibility violin plots were computed using the GeneActivity function. Peaks were linked to gene expression and visualized using CoveragePlot.

**Tissue fixation, processing, and sectioning**. Morphologically healthy antral follicles and surrounding tissue were resected from the surface of ovaries obtained from tissue donors. Tissue was washed using cold DPBS and submerged in 2 mL of 4% Paraformaldehyde (PFA, Thermo Scientific). Tissue in 4%PFA was fixed overnight in 4 °C, followed by dehydration in 2 mL of 30% Sucrose (Sigma) overnight at 4 °C. Tissue was embedded in cryomold with OCT (TissueTek) and rapidly frozen by placing on a metal mold cooled using liquid nitrogen. Tissue was kept at −80 °C until sectioned. Cryosections of 10 μm thickness were obtained using the Cryostat CM 3050 S (Leica) and stored at −20 °C prior to protein immunofluorescence or RNA ISH.

**Protein immunofluorescence**. Cryosections were labeled as previously described[59,60]. Samples were permeabilized in PBS/0.1% TWEEN (PBST) with 5% donkey serum (Millipore) and incubated in 2–5 μg/ml concentrations of specified antibodies at room temperature for 1 h, washed and counterstained with DAPI and mounted in Prolong Gold (Gibco). If antibodies were not pre-conjugated, primary incubation was performed overnight at 4 °C, followed by incubation with secondary antibody (concentration of 1:250) at room temperature for an hour. For all immunolabeling experiments, fluorophore-conjugated secondary antibody alone was used as a negative control Images were captured using a Zeiss 710 Confocal Microscope.

**RNA In situ hybridization (RNAScope)**. Human ovarian PFA fixed frozen tissue was mounted on glass slides and stained on an automated staining platform (Leica Bond RX) using the RNAScope 2.5 LS Assay Reagent Kit-Red (Advanced Cell Diagnostics) as recommended by the manufacturer. Tissue was stained for the target probes Hs-HHIP, Hs-PTCH1, Hs-TCF21, Hs-NR5A1 and staining was validated by also staining with a positive control probe against Hs-PPIB and the Negative Control Probe-dapB (all probes were obtained from Advanced Cell Diagnostics). Positive RNA hybridization was identified as punctate red dots using the red channel on a confocal microscope (Zeiss 710 Confocal Microscope).

**Procurement of ovarian tissue**. Ovaries were processed as previously described in ref. [61]. Briefly, cortical tissue was separated from medulla and further processed to remove medulla tissue and thinned cortex was cut into fragments of roughly 3 × 12 mm. Each cortical strip was then equilibrated in cryoprotectant solution in cryovials, slow-frozen and stored in liquid nitrogen until thaw and transplantation. All tissue was obtained from donors and patients with informed consent and with approval of the Weill Cornell Institutional Review Board— Study title: Low temperature preservation of ovarian tissue; Study number: 0803009702.

**Endothelial cells**. Human ECs were obtained from Angiocrine Bioscience and were originally isolated from neonatal Umbilical Vein (HUVEC) as described[62] under an IRB approved protocol for use of discard biological material. HUVEC were isolated and expanded for three passages in endothelial cell growth medium before cryopreservation.

**Ovarian cortical xenografts**. All procedures were approved, and experiments were performed in accordance with the guidelines and regulations of the Institutional Animal Care and Use Committee (IACUC) of Weill Cornell Medicine (IACUC Protocol #2014-0008—Assessment of angiogenic and hematopoietic tissue in mouse). These procedures were previously described in refs. [59,60]. Briefly, cryopreserved tissue was thawed rapidly, washed of cryoprotectant, and encapsulated in fibrin that was pre-mixed with labeled single-cell suspension of ECs. Fibrin embedded tissue was then transplanted into two oophorectomized immunocompromised 10 week-old female NOD SCID mice (NSG) mice bilaterally under the fascia of the Gluteus Maximus and the fascia, dorsal wall skin closed with sutures. Xenografted animals were maintained in sterile conditions until animals were euthanized and xenografts recovered for fixation, cryosectioning, and immunohistochemical staining.

**Ethynyl deoxyuridine (EdU) and 5-chloro-2′-deoxyuridine (CldU) injections**. Two NSG mice were bilaterally transplanted with human ovarian cortical pieces in a fibrin matrix with ECs (n = 4). After 11 weeks when antral follicle growth was observed, mice were sequentially injected with Edu (Sigma, 100 mg/kg for 2 days) and then CldU (Sigma, 100 mg/kg for 2 days) 24 h apart. A day later, mice were sacrificed, and the xenografts were recovered, the tissue was fixed in 4% PFA overnight followed by dehydration in 30% Sucrose overnight at 4 °C. The tissue was embedded in OCT (TissueTek) and cryosectioned for labeling of EdU and CldU incorporation.

**Ethynyl deoxyuridine (EdU) and 5-chloro-2′-deoxyuridine (CldU) staining**. For double staining of slides for EdU and CldU protocol was modified from Podgorny et al.[63]. Cryosections were permeabilized with PBS/0.1% TWEEN (PBST) with 5% donkey serum (Millipore) for 1 h, followed by 3× washes with PBS. For EdU staining the Click-it Edu Cell Proliferation Kit for Imaging, Alexa Fluor 555 dye (Invitrogen) was used according to manufacturer's instructions, followed by 3× washes in PBS and a second click reaction with 2 mM azidomethyl phenyl sulfide (Sigma), 20 mM (+)-sodium L-ascorbate (Invitrogen), 4 mM Copper Sulfate (Invitrogen) in PBS for 15 min. After washing with PBS, the slides were denatured with 2 N HCl (Thermo Scientific) for 30 min and quickly neutralized with two 10 min 0.1 M Borate washes (Thermo Scientific) incubations. Slides were washed with PBS and blocked with PBS/0.1% TWEEN (PBST) with 5% donkey serum (Millipore) for 1 h followed by incubation with rat anti-BrdU antibody (Abcam) overnight at 4 °C. The following day cryosections were incubated with secondary antibody for 1 h, washed 3× with PBS, counterstained with DAPI and mounted in Prolong Gold (Gibco). Images were captured using a Zeiss 710 Confocal Microscope.

**Ethynyl deoxyuridine (EdU) and 5-chloro-2′-deoxyuridine (CldU) quantification**. For quantification of EdU and CldU incorporation (Fig. 4) in early follicles (primordial, primary, and secondary <5 GC layers) boundaries were manually drawn on each follicle (n = 45) to obtain measurements for either the GC layer or the theca layer immediately surrounding the follicle. Using the Zen colocalization software the %Edu+ or CldU+ area for each region was quantified by dividing the area of colocalization of Edu or CldU with nuclear staining (DAPI+) by the overall nuclear area (DAPI+). Area of EdU and CldU double staining was calculated by dividing the area of overlap by the total area of staining for both dyes. Only follicles with proliferating cells were considered for analysis. Statistical analyses and graphs were done using Prism 9 (Graphpad), significance was calculated using multiple unpaired t-tests.

**Quantification of Ethynyl deoxyuridine (EdU), ki67 and CYP17A1/ANPEP/THBD staining**. As previously described cryosections were stained for EdU, followed by staining for ki67 and the specific cell marker of interest (CYP17A1, ANPEP or THBD). Six non-consecutive cryosections from a 3 mm follicle were stained and quantified using the Zen colocalization software. Five regions from each image were randomly chosen for quantification, and number of proliferative cells was calculated by manually counting the number of cells expressing the marker of interest (CYP17A1/ANPEP/THBD) and the proliferation marker (either Ki67 or EdU) and dividing that number over the total number of cells expressing the marker of interest. For example: Number of ANPEP+EdU+ cells/Number of ANPEP+ cells. Statistical analyses and graphs were done using Prism 9 (Graphpad), significance was calculated using multiple unpaired t-tests.

**Statistics and reproducibility**. A p value of <0.05 was considered significant for all experiments, however values determined by statistical analysis of differentially expressed genes in scRNASeq data were much smaller. To illustrate the degree of significance of these differences, the p values for scRNASeq analysis are specified. For significance of cell proliferation (EdU/CldU incorporation Ki67 in Fig. 4), unpaired two-sided T-test was used to calculate p values. For scRNASeq and scATACSeq data, the p values for each comparison were calculated using the Wilcoxon rank-sum test.

**Antibodies**. Table 1.

**Reporting summary**. Further information on research design is available in the Nature Portfolio Reporting Summary linked to this article.

## Data availability

All data needed to evaluate the conclusions in this paper are present in the paper and/or the Supplementary Materials. Source data for Fig. 1a–c, Fig. 2a, Supp Fig. 1a–f, and Supp Fig. 2 are contained in Supplementary Data 1. Source data for Fig. 5a–f and Supp Fig. 3a, b are contained within the snRNA and scATAC-seq libraries, located on GEO, record number GSE192722. Single cell multi-omics libraries, as well as scripts and

**Table 1 Antibodies.**

| Antibody name | Company | Catalogue #. |
|---|---|---|
| Alexa Fluor 488 anti-human CD45 (clone HI30) | BioLegend | Cat#304019 |
| DAPI | Invitrogen | Cat#D1306 |
| PE anti-human (CD13) ANPEP (clone WM15) | BioLegend | Cat#301703 |
| APC anti-human (CD13) ANPEP (clone WM15) | eBiosciences | Cat# 17-0138-41 |
| PE anti-human CD31 (clone WM59) | BioLegend | Cat#303106; |
| FITC anti-human CD55 (clone JS11) | BioLegend | Cat#311306; RRID:AB_314863 |
| FITC anti-human CD99 (clone 3B2/TA8) | BioLegend | Cat#371303 |
| APC anti-human CD99 (clone 3B2/TA8) | BioLegend | Cat#371308 |
| TCF21 Polyclonal Antibody | Invitrogen | Cat# PA5-53031 |
| Human/Mouse/Rat alpha- Smooth Muscle Actin Antibody | R&D Systems | Cat# MAB1420 |
| Alexa Fluor 647 anti-human CD141(Thrombomodulin)—Clone (M80) | BioLegend | Cat# 344123 |
| Gli1 Polyclonal Antibody | Invitrogen | Cat# PA5-72967 |
| Human/Mouse Patched 1/PTCH (First Extracellular Loop) Antibody | R&D Systems | Cat# MAB41051 |
| CYP11A1 (D8F4F) Rabbit mAb | Cell Signaling Technologies | Cat# 14217 S |
| Human Endoglin/CD105 Alexa Fluor 488-conjugated Antibody | R&D Systems | Cat# FAB10971G |
| CYP17A1 (E6A7G) XP Rabbit mAb | Cell Signaling Technologies | Cat# 94004 S |
| PE anti-human Podoplanin (PDPN) Antibody | BioLegend | Cat# 337003 |
| Human CD34 APC-conjugated Antibody | R&D Systems | Cat# FAB7227A |
| Anti-BrdU antibody [BUI/ 75 (ICR1)] | Abcam | Cat# ab6326 |
| Pe anti-human CD141(Thrombomodulin)—Clone (M80) | BioLegend | Cat# 344103 |
| Anti-Ki67 antibody | Abcam | Cat #ab15580 |
| Anti-CYP17 Antibody (D-12) | Santa Cruz | Cat# sc-374244 |
| Alexa Fluor Plus 647 goat anti-rabbit | Invitrogen | Cat# A32733 |
| Alexa Fluor 647 goat anti-rat | Invitrogen | Cat# A21247 |
| Alexa Fluor 488 donkey anti-rabbit IgG | Life Technologies | Cat# A21206 |
| Alexa Fluor 488 goat anti-rat | Invitrogen | Cat# A11006 |
| Alexa Fluor 555 donkey anti-rabbit IgG | Life Technologies | Cat# A31572 |
| Anti-hBiglycan affinity purified Goat IgG | R&D Systems | Cat# AF 2667 |
| PE anti-CD105 (Endoglin) | Biolegend | Cat# 800503 |
| Alexa Fluor 488 donkey anti-goat IgG | Life technologies | Cat# A11055 |

metadata are available on GEO, record number GSE192722. Source data for Fig. 4d–f and h are contained in Supplementary Data 2.

## Code availability

Code associated with the analysis of data is posted along with multi-omics libraries on GEO, record number GSE192722.

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

## Acknowledgements
The authors acknowledge support from the Queenie Victorina Neri Research Scholar Award. N.L.G is supported by the NYSTEM Stem Cell and Regenerative Medicine postdoctoral training grant. The authors also acknowledge the assistance provided by the Genomic Resources Core Facility at Weill Cornell Medicine for single-cell RNA-Sequencing, snRNA Sequencing and scATAC sequencing and alignment; special thanks go to Aihong Liu, Adrian Tan, Xing Wang, and Jenny Xiang. The authors acknowledge the assistance of the Laboratory for Comparative Pathology at Memorial Sloan Kettering Cancer Center for RNA in situ hybridization staining; special thanks to Sebastien Monette.

## Author contributions
Conceptualization, N.L.G. and D.J.; Methodology, N.L.G., E.K., J.W., L.A., A.T.K., G.G., K.Z., S.J., R.B., and D.J.; Resources, N.Z., N.S.S., G.S., and Z.R.; Software, N.L.G., J.W., and D.J.; Formal analysis, N.L.G., L.M., and D.J.; Writing—Original Draft, N.L.G. and D.J.; Writing—Review & Editing, N.L.G., L.M., and D.J.; Supervision, Z.R. and D.J.

## Competing interests
The authors declare no competing interests.
