## [Peer Review File · Communications Biology]

Reviewers' comments:

Reviewer #1 (Remarks to the Author):

Dear authors

The manuscript "the theca cell layer of human follicles derives from ovarian stromal cells and is comprised of three lineage specific subtypes" provide transcriptional evidence (and some validation by immunostaining) that in human small antral follicles there seem to be 3 theca cell types (structural, androgenic and perifollicular) instead of the text-book two types (interna/androgenic and externa/structural). This paper has been previously submitted and although I consider that it has improved in quality, many aspects of the presentation remain poor/unclear, the bioinformatics part (number of clusters seem exaggerated, suggesting that the parameters were not well chosen, so that the DEGS etc are less meaningful)

The lines mentioned in this document refer to the "marked manuscript" received.

Figure 1 is confusing and unclear. If the authors separated the theca population and granulosa population per antral follicle first enzymatically (Fig 1C) followed by FACS sorting (Fig 1D and 1E) how come you then provide a UMAP containing SMC, EC, Hem, OE and uncharacterized clusters? This needs to be clarified. Reading the Material and Methods, Figure 1A-E is not informative or connected to Figure 1F and should be removed or transferred to a Figure where it actually links to the technique used (such as Figure 6).

Line 88: It is unclear how do the authors reach the conclusion that there are 6 main clusters. What is this based on? If I look at Figure 1G I don't see how the conclusion was reached? In other words what is the evidence that the 22 clusters identified belong to 6 main clusters?

Line 90: cluster "theca" is presumably "theca/stroma"? Otherwise where is the stromal compartment? This needs to be clarified and standardized (see Figure 6C and 7A, that present "theca/stroma" main clusters).

Line 90: Can the authors please analyse the remaining single cell population and assign clusters + validate? At least attempt to analyse the remaining cells at the center of Figure 1g? You cannot just simply ignore the cells at the center of Figure 1F.

Line 94: can the authors mention in the main text the clusters what were the "theca" clusters used for subclustering in Figure 2 explicitly? In my opinion cluster 20 should be included in the main-theca cluster (in Figure 1G shows high expression of CYP11 and CYP17; those are androgenic/interna theca!). Why was this cluster not included as main-theca (in Figure 1F)? If the main-theca cluster includes the stroma and theca and was used for sub-clustering, what cells are then stroma in Figure 2? This needs to be clarified, also in the characterization that follows in Figure 3 (there should be where is the stroma, compare to Figure 6C and 7A).

Line 97-98: I don't understand the sentence "for libraries that contained GCs...". Are the authors trying to identify theca cells from follicles in atresia from the 13 follicles analysed? This place in the text to mention this is now out of context. Can this analysis (atresia) be included in Figure 1 and mentioned in the previous paragraph, concluding that follicle Pt2_1 is undergoing atresia.

Line 260-261: "dotted lines in Figure 2A? Figure 2A is a UMAP. Please correct this mistake.

Figure 3: the information provided in Figure 2C is repeated in Figure 3A. Please substitute Figure 2C by 3A. You should not repeat the same information in 2 different Figures in the same paper.

From Figure 2 it is unclear how many populations of theca there are, perhaps replacing Figure 2A by 3A, and using the dashed lines it is clear that you are referring to 4 different theca populations. As many of your clusters are very similar and that is purely determined by the chosen resolution in Seurat, it remains unclear why you don't adapt the resolution in Seurat to your biologically different clusters.

How sure are you that the Cluster 7 in Figure 3A is a bonafide sub-population? Please validate this population of perhaps you can change the resolution in Seurat.

Figure 3 and 4: The immunostaining for TCF21 and PTCH1 is not specific for the designated theca layer. The use of these antibodies presumably giving aspecific staining should be avoided.

Figure 5A: Could the DAPI channel be merged with the KI67 channel (DAPI+KI67) to appreciate that the Ki67 staining is indeed nuclear.

Figure 6: You purified 4 FACS-sorted theca populations and cultured for 36h. Were you trying to differentiate the cells further? If so into what? The conclusion that pTcCs are a "terminal fate" is premature if the culture period was only 36h and no directed differentiation was attempted. The text should be changed to state that the pTcC can not differentiate into aTcC, when exposed to the medium used for 36h.

I find it confusing that the authors refer to 4 main theca subpopulations throughout the text, but in the title refer to 3 subpopulations? This should be clarified. But now I notice that the title on the marked version provided is not the same as in the clean version.

Reviewer #2 (Remarks to the Author):

The theca cell layer of human follicles derives from ovarian stromal cells and is comprised of three lineage specific subtypes. Lustgarten Guahmich et al.

In this study, Prof. Rosenwaks group used several tools to identify and trace the differentiation path of different subgroups of theca cells. They demonstrated that the theca compartment is much more complex than what we have learned in the textbooks. Although we already know through in vitro studies with bovine and human ovarian stromal cells (Orisaka et al., Biol Reprod 2006; Asiabi et al., Hum Reprod 2020) that theca cells originate from them, this study indicates that three different cell subtypes give rise to the structural, androgenic and perifollicular theca cells, being the latter one never described before. The authors showed that cell recruitment to form the theca layers starts very early, during the activation of primordial follicles, confirming the crucial role of the follicles on theca cell differentiation. Through factors synthesized by granulosa cells and the oocyte, the follicles coordinate the proliferation, recruitment and differentiation of cells surrounding them. However, this process remains elusive, as well as the origin of these cell subtypes. Are they specific subgroups of stromal cells to begin with, or just stromal cells that happen to be in the right place at the right time? Unfortunately, there is still no answer to this question.

While this is a very interesting study, my main concern is the limited number of samples used in some experiments. For instance, ovarian tissue samples from only two patients were used for RNA in situ hybridization. I believe the authors should mention this and the reasons for caution in their discussion. Moreover, the xenotransplantation experiment was performed using ovarian tissue from only one donor. I do not think this is reliable enough to draw conclusions on the onset of theca cell differentiation during follicle activation. Same for the in vitro culture experiment.

General comments:

1) Positive and negative controls for the immunofluorescence staining are missing. Please mention

them if they were used. If they were not used, the staining should be repeated with the proper controls.

2) Some immunofluorescence staining pictures in Figure 4 are hard to identify the positive cell pointed by the arrows. Please replace them with another picture or add a higher magnification.

Specific comments:

- L98: reference #3 is not correct, as this study did not isolate theca cells from antral follicles.
- Some typos (e.g., THMD instead of THBD) can be found in the text.

Reviewer #3 (Remarks to the Author):

The manuscript "Specification and lineage diversification of theca cells in human ovarian follicles is governed by a TCF21/NR5A1 axis" describes theca cell populations in growing follicles of the human ovary. The authors propose a differentiation trajectory for theca cells after follicle activation and describe theca-cell subpopulations in antral follicles.

This manuscript provides valuable information to the field as ovarian theca cells are poorly described. Overall, the manuscript is well written and provides enough evidence to support the findings.

I have some minor comments:

- 1) Under Data Availability it's written that single cell-multi-omics data is available on GEO. Please add accession numbers there.
- 2) In figure 2 legend it's written that the p-value for significantly enriched genes is $< 1 \times 10^{-72}$. In figure 3 legend the p-value is $< 1 \times 10^{-40}$. What is the justification to use different cut-offs for differentially expressed genes? Moreover, I advise adding more information to the Statistical analyses section in Materials and Methods (lines 1039-1041). A p-value of less than .05 was considered significant in what type of experiments?
- 3) I advise using one format in Materials and Methods from start to end. For example, at the moment there's 1 ml or 1ml, 4C or 4°C, 1h or 1 hour.
- 4) Supplementary Table 2 "All cells" sheet is missing one column name.

Reviewer #1:

Dear authors

The manuscript "the theca cell layer of human follicles derives from ovarian stromal cells and is comprised of three lineage specific subtypes" provide transcriptional evidence (and some validation by immunostaining) that in human small antral follicles there seem to be 3 theca cell types (structural, androgenic and perifollicular) instead of the text-book two types (interna/androgenic and externa/structural). This paper has been previously submitted and although I consider that it has improved in quality, many aspects of the presentation remain poor/unclear, the bioinformatics part (number of clusters seem exaggerated, suggesting that the parameters were not well chosen, so that the DEGS etc are less meaningful) The lines mentioned in this document refer to the "marked manuscript" received.

Comment #1:

Figure 1 is confusing and unclear. If the authors separated the theca population and granulosa population per antral follicle first enzymatically (Fig 1C) followed by FACS sorting (Fig 1D and 1E) how come you then provide a UMAP containing SMC, EC, Hem, OE and uncharacterized clusters? This needs to be clarified. Reading the Material and Methods, Figure 1A-E is not informative or connected to Figure 1F and should be removed or transferred to a Figure where it actually links to the technique used (such as Figure 6).

We acknowledge that our protocols for isolation of cells from antral follicles does not produce a pure population of granulosa and/or theca cells and we have modified the figure presentation as this Reviewer suggests to clarify the disposition of cells. Although the described method for sorting cells highly enriches for populations of interest (GCs and TCs), sorting of the CD55 fraction is inclusive of other cell populations that are present in ovarian tissue fragments containing antral follicles. This includes endothelial cells (ECs), vascular-associated smooth muscle cells (SMCs), hematopoietic cells (Hem), and ovarian epithelium (OE). Fig1A-E is intended to define the parameters by which cells were enriched, however we agree that these data are not critical and are also previously published by our group, hence we have removed them. To clarify the presentation of our data, we have also moved Fig1F-G to the Supplement, where the rationale for selection of Seurat clusters 1, 2, 3, 4, 5, 11, and 18 as theca cells is justified.

Comment #2:

Line 88: It is unclear how do the authors reach the conclusion that there are 6 main clusters. What is this based on? If I look at Figure 1G I don't see how the conclusion was reached? In other words what is the evidence that the 22 clusters identified belong to 6 main clusters?

Cells were allocated to the 6 main clusters using canonical, published markers for each of the specified cell types, however many clusters were derived from cells with low features and/or counts. To clearly identify these clusters, we have added asterisks to the violin plots representing these clusters (previously Fig 1G, now Supplemental Fig 1C-E). Low features/counts suggests a high number of unassigned reads and cells of poor quality. For these reasons and to utilize only high-quality cells in our downstream analysis, we have excluded these clusters from our curated Theca/Stroma analysis.

Comment #3:

Line 90: cluster “theca” is presumably “theca/stroma”? Otherwise where is the stromal compartment? This needs to be clarified and standardized (see Figure 6C and 7A, that present “theca/stroma” main clusters).

We appreciate the Reviewer pointing out this and other inconsistencies. We will change wording where appropriate to “theca/stroma” throughout the text.

Comment #4:

Line 90: Can the authors please analyse the remaining single cell population and assign clusters + validate? At least attempt to analyse the remaining cells at the center of Figure 1g? You cannot just simply ignore the cells at the center of Figure 1F.

As described above in response to Comment #2, clusters represented in the middle of the UMAP plot revealed a low number of Features/Counts reflecting poor quality of the cells contained within. We have shown evidence justifying exclusion of these clusters in Supplementary figure 1. We acknowledge that diminished sample quality is a shortfall of the tissue processing, especially for theca/stroma cells which are challenging to obtain and require enzymatic and mechanic processing. Nevertheless, based on our own previously published datasets, as well as analysis of cell libraries generated by other studies (e.g. Fan et al., previous response to reviewers), we have concluded that the cell types retained for downstream analysis are representative of theca/stroma cells in antral follicles that have previously been analyzed by scRNASeq.

Comment #5:

Line 94: can the authors mention in the main text the clusters what were the “theca” clusters used for subclustering in Figure 2 explicitly? In my opinion cluster 20 should be included in the main-theca cluster (in Figure 1G shows high expression of CYP11 and CYP17; those are androgenic/interna theca!). Why was this cluster not included as main-theca (in Figure 1F)? If the main-theca cluster includes the stroma and theca and was used for sub-clustering, what cells are then stroma in Figure 2? This needs to be clarified, also in the characterization that follows in Figure 3 (there should be where is the stroma, compare to Figure 6C and 7A).

Thank you to the reviewer for pointing this out. Although cluster 20 does show a similar expression profile to the other cluster identified as androgenic theca (#18), it is excluded because QC demonstrated that this cluster is of poor quality (low Features/Counts, see response to Comments#2/4). Regarding the allocation of stroma, we have revised our designation throughout the manuscript to reflect that mixed cell populations are comprised of both theca and stroma cells.

Comment #6:

Line 97-98: I don't understand the sentence “for libraries that contained GCs...”. Are the authors trying to identify theca cells from follicles in atresia from the 13 follicles analysed? This place in the text to mention this is now out of context. Can this analysis (atresia) be included in Figure 1 and mentioned in the previous paragraph, concluding that follicle Pt2_1 is undergoing atresia.

We agree that the wording and placement of this sentence is confusing. To clarify, for some library preparations, purified GCs and theca/stroma were combined for economy and so that GCs could provide an indication of the overall “health” of the follicle, as GC atresia markers have been widely validated and published. We will change the wording to

“Among follicles for which corresponding GCs were also sequenced”. In regard to the movement of the atretic marker analysis, we have added this file to Supplementary Data related to Figure 1 (Supp Fig 1F).

Comment #7:

Line 260-261: “dotted lines in Figure 2A? Figure 2A is a UMAP. Please correct this mistake. **The dotted lines we refer to here are the stroked lines that group “closely associated clusters”. This now refers to Figure 1A.**

Comment #8:

Figure 3: the information provided in Figure 2C is repeated in Figure 3A. Please substitute Figure 2C by 3A. You should not repeat information in 2 different Figures in the same paper. **We have merged the data from 2B into 3A while retaining the purpose of the data by using UMAP feature plots to highlight the markers shown in Figure 2D-G.**

Comment #9:

From Figure 2 it is unclear how many populations of theca there are, perhaps replacing Figure 2A by 3A, and using the dashed lines it is clear that you are referring to 4 different theca populations. As many of your clusters are very similar and that is purely determined by the chosen resolution in Seurat, it remains unclear why you don't adapt the resolution in Seurat to your biologically different clusters.

The resolution that we have used for the FindClusters function in Seurat is the default (0.8) setting. While we appreciate the Reviewer's suggestion, and have attempted lower resolution settings to simplify the cluster designations, we have found that reduced resolution results in failure of DEG analysis (FindAllMarkers function in Seurat) to highlight transcripts (e.g. PTCH1, THBD) that we independently corroborate by InSitu analysis and immunolabeling to be present in the theca/stroma layer of antral follicles. Indeed, while scRNASeq provides unprecedented resolution, it is as the Reviewer stated – clustering is arbitrarily designated, and populations must be discriminated based on their biological relevance. In this case, we have used the default resolution and unbiased analysis across multiple follicles of different stages to highlight transcripts that are significantly enriched across cell groups. We have then corroborated the presence/absence of these markers and defined their localization across follicle development using additional means.

Comment #10:

How sure are you that the Cluster 7 in Figure 3A is a bonafide sub-population? Please validate this population or perhaps you can change the resolution in Seurat.

As stated in response to Comment #9, we have used the default resolution to identify transcripts that are significantly enriched across clusters. However, we did not intend for this cluster to be designated as a unique sub-population, but rather as a cluster with significant enrichment of PTCH1 and BGN and lacking significant enrichment of THBD and/or HHIP. While the cluster resolution is tunable, our intent is to provide a framework whereby single-cell UMAP plots and Seurat analysis can be overlaid with a visual (InSitu or immunolabeling) approach. While we do highlight a lineage-negative progenitor that may align with this 4th population of PTCH1⁺BGN⁺THBD^{neg}HHIP^{neg} cells in Figure 3A, we have clarified in the revision that this is a putative cell identity that is not captured within the scRNASeq analysis of antral stage follicles [“Importantly, while Lin^{neg}TPCs could not

be specifically identified here based on expression of surface markers, we have designated these as a putative cell that resides between the aTC and sTC layer, in THBD^{neg}ANPEP^{neg}ACTA2^{neg} cells that express low *TCF21* (Fig 1d), high *PTCH1* (Fig 1e), and no *HHIP* (Fig 1f)"].

Comment #11:

Figure 3 and 4: The immunostaining for TCF21 and PTCH1 is not specific for the designated theca layer. The use of these antibodies presumably giving aspecific staining should be avoided.

We acknowledge that the antibodies have background staining and it is for this reason that we have used RNA immunostaining as corroboration of the expressed markers.

Comment #12:

Figure 5A: Could the DAPI channel be merged with the Ki67 channel (DAPI+Ki67) to appreciate that the Ki67 staining is indeed nuclear.

We have utilized green arrows to designate Ki67⁺ cells and these arrows are present in all fields, thereby demonstrating nuclear localization.

Comment #13:

Figure 6: You purified 4 FACS-sorted theca populations and cultured for 36h. Were you trying to differentiate the cells further? If so into what? The conclusion that pTTCs are a "terminal fate" is premature if the culture period was only 36h and no directed differentiation was attempted. The text should be changed to state that the pTTC can not differentiate into aTC, when exposed to the medium used for 36h.

We acknowledge that the in vitro studies are limited given the shorter time course. We decided on this time-period given a previous publication that showed significant differentiation of stromal cells into androgen producing cells after 48 hours of culture (<https://pubmed.ncbi.nlm.nih.gov/33326997/>); based on this, we hypothesized that a shorter time-period might provide insight into intermediate populations. Nevertheless, our conclusions based on these data are premature as the Reviewer suggested. Additionally, following from concerns that this experiment utilized cells isolated from multiple follicles of a single patient, we have removed these data from the manuscript, as they are no longer supportive of downstream experimental goals.

Comment #14:

I find it confusing that the authors refer to 4 main theca subpopulations throughout the text, but in the title refer to 3 subpopulations? This should be clarified. But now I notice that the title on the marked version provided is not the same as in the clean version.

Our intent in our study is to provide a rigorous account of theca cell specification and differentiation and we aim to show that there are 3 phenotypically unique theca derivatives present in antral follicles that arise from a 4th progenitor cell. We will clarify that the progenitor cell that we have inferred has not been clearly characterized in our analysis and phenotypic and functional potential of these cells is yet to be demonstrated.

Reviewer #2:

The theca cell layer of human follicles derives from ovarian stromal cells and is comprised of three lineage specific subtypes. Lustgarten Guahmich et al.

In this study, Prof. Rosenwaks group used several tools to identify and trace the differentiation path of different subgroups of theca cells. They demonstrated that the theca compartment is much more complex than what we have learned in the textbooks. Although we already know through in vitro studies with bovine and human ovarian stromal cells (Orisaka et al., Biol Reprod 2006; Asiabi et al., Hum Reprod 2020) that theca cells originate from them, this study indicates that three different cell subtypes give rise to the structural, androgenic and perifollicular theca cells, being the latter one never described before. The authors showed that cell recruitment to form the theca layers starts very early, during the activation of primordial follicles, confirming the crucial role of the follicles on theca cell differentiation. Through factors synthesized by granulosa cells and the oocyte, the follicles coordinate the proliferation, recruitment and differentiation of cells surrounding them. However, this process remains elusive, as well as the origin of these cell subtypes. Are they specific subgroups of stromal cells to begin with, or just stromal cells that happen to be in the right place at the right time? Unfortunately, there is still no answer to this question.

While this is a very interesting study, my main concern is the limited number of samples used in some experiments. For instance, ovarian tissue samples from only two patients were used for RNA in situ hybridization. I believe the authors should mention this and the reasons for caution in their discussion. Moreover, the xenotransplantation experiment was performed using ovarian tissue from only one donor. I do not think this is reliable enough to draw conclusions on the onset of theca cell differentiation during follicle activation. Same for the in vitro culture experiment.

We thank this Reviewer for their careful assessment of our manuscript and for their appreciation of the significant knowledge gaps in understanding of follicle development. We also acknowledge that some of our experiments were limited by the scarcity of human ovarian tissue and high expense of the analyses we have conducted. Specifically, we recognize that single-cell RNASeq analysis of phenotypic sub-types that were isolated from multiple follicles and cultured for 36 hours (Figure 6), while illustrative of the differentiation potential of each sub-type, represents a single biological replicate. To ensure the rigor of our study we have agreed to exclude these data from the manuscript. Regarding the two other concerns related to replicates, we respectfully disagree and contend that the rigor of these data are appropriate for the following reasons:

- 1) In Situ Hybridization data - Patterns of expression were observed across multiple (>5 antral follicles; >50 primordial, primary secondary and pre-antral) follicles representing 2 patients. While the Reviewer concerns relate to n=2, we argue that the descriptive data we generated is generally representative of follicles across these two patients, and more importantly, the data are corroborated by transcriptomic evidence from our own and other groups as well as immunolabeling approaches that we utilize later in the manuscript. Based on the totality of this evidence we believe that our InSitu data is appropriate as it is.**
- 2) EdU/CldU incorporation – The Reviewers expressed concerns that these data derived from n=1 patient. While this is true, we emphasize that ovarian tissue from that single patient was distributed across n=4 xenografted cortical fragments. While the individual biology of a patient must always be taken into consideration as a variable,**

we argue that in this case genetic background of ovarian tissue is less relevant, with each cortical fragment representing a bona fide biological replicate. Moreover, the observations of increased proliferation within the THBD⁺ sub-population is evident in the transcriptomic data across multiple donors and trends in proliferation revealed by EdU/CldU incorporation are also illustrated by measurement of Ki67. Taken together, we argue that these data provide sufficient support for our claims. Nevertheless, given the limited replicate number, we have incorporated language into the revision that highlights this as a limitation to be considered when assessing our study.

General comments:

1) Positive and negative controls for the immunofluorescence staining are missing. Please mention them if they were used. If they were not used, the staining should be repeated with the proper controls.

We apologize for excluding this detail from our Materials and Methods. It is routine practice for our group to use positive and negative controls with our immunolabeling and InSitu Hybridization analysis every time the procedures are performed.

2) Some immunofluorescence staining pictures in Figure 4 are hard to identify the positive cell pointed by the arrows. Please replace them with another picture or add a higher magnification. **Thank you to the reviewer for pointing this out, we will add higher magnification images.**

Specific comments:

- L98: reference #3 is not correct, as this study did not isolate theca cells from antral follicles.

We acknowledge that we misidentified the methods used in this study. We have revised the text to read, "...mechanical dissociation of the theca layer from isolated antral follicles or derivation from isolated stromal cells" and will attribute the work separately.

- Some typos (e.g., THMD instead of THBD) can be found in the text.

Thank you for pointing this out, they will be corrected in the revision.

Reviewer #3:

The manuscript “Specification and lineage diversification of theca cells in human ovarian follicles is governed by a TCF21/NR5A1 axis” describes theca cell populations in growing follicles of the human ovary. The authors propose a differentiation trajectory for theca cells after follicle activation and describe theca-cell subpopulations in antral follicles.

This manuscript provides valuable information to the field as ovarian theca cells are poorly described. Overall, the manuscript is well written and provides enough evidence to support the findings.

I have some minor comments:

1) Under Data Availability it's written that single cell-multi-omics data is available on GEO. Please add accession numbers there.

The accession number is GSE192722 and will be added to the finalized manuscript.

2) In figure 2 legend it's written that the p-value for significantly enriched genes is $< 1 \times 10^{-72}$. In figure 3 legend the p-value is $< 1 \times 10^{-40}$. What is the justification to use different cut-offs for differentially expressed genes? Moreover, I advise adding more information to the Statistical analyses section in Materials and Methods (lines 1039-1041). A p-value of less than .05 was considered significant in what type of experiments?

The materials and methods have been modified as follows: “A p-value of less than .05 was considered significant for the EdU/CldU quantification experiments (Fig 5).”

3) I advise using one format in Materials and Methods from start to end. For example, at the moment there's 1 ml or 1ml, 4C or 4°C, 1h or 1 hour.

We apologize for the poor formatting and will standardize the unit forms in the revision.

4) Supplementary Table 2 “All cells” sheet is missing one column name.

We apologize for including these values as they were intended for our own analysis. They have been removed from the revised Table.

REVIEWERS' COMMENTS:

Reviewer #1 (Remarks to the Author):

Dear Authors,

You have taken on board some of the suggestions and the manuscript has increased in quality and clarity.

Reviewer #2 (Remarks to the Author):

I have no further comments.